# Dietary and Genetic Cholesterol Loading Rather Than Steatosis Promotes Liver Tumorigenesis and NASH-Driven HCC

**DOI:** 10.3390/cancers13164091

**Published:** 2021-08-13

**Authors:** Vicent Ribas, Laura Conde de la Rosa, David Robles, Susana Núñez, Paula Segalés, Naroa Insausti-Urkia, Estel Solsona-Vilarrasa, José C. Fernández-Checa, Carmen García-Ruiz

**Affiliations:** 1Department of Cell Death and Proliferation, Institute of Biomedical Research of Barcelona (IIBB), CSIC, 08036 Barcelona, Spain; vribas@clinic.cat (V.R.); lconder@clinic.cat (L.C.d.l.R.); david.robles@iibb.csic.es (D.R.); snunez@clinic.cat (S.N.); paula.segales@iibb.csic.es (P.S.); naroa.insausti@iibb.csic.es (N.I.-U.); estel.solsona@iibb.csic.es (E.S.-V.); 2Liver Unit, Hospital Clinic I Provincial de Barcelona, Instituto de Investigaciones Biomédicas August Pi i Sunyer (IDIBAPS), 08036 Barcelona, Spain; 3Center for the Study of Liver and Gastrointestinal Diseases (CIBERehd), Carlos III National Institute of Health, 28029 Madrid, Spain; 4Center Esther Koplowitz, Planta Cuarta, C/Rosselló 149, 08036 Barcelona, Spain; 5Center for ALPD, Keck School of Medicine, University of Southern California, Los Angeles, CA 90089, USA

**Keywords:** cholesterol, hepatocellular carcinoma, steatosis, NASH

## Abstract

**Simple Summary:**

In the present study, which is part of the Special Issue “Theranostic Advances in Hepatobiliary Tumors”, we address whether hepatic steatosis per se or cholesterol sensitizes to NASH-driven HCC. This is a very important health issue, as the incidence of HCC derived from NASH is expected to keep rising due to the association of NASH with the obesity and type 2 diabetes epidemic. Using dietary and genetic models to generate hepatic steatosis with or without cholesterol accumulation, we provide evidence for the tumor promoter role of cholesterol in NASH-HCC associated with an increased expression of the genes involved in immune checkpoints, which suggests that cholesterol favors a milieu prone to T-cell exhaustion.

**Abstract:**

The association of nonalcoholic steatohepatitis (NASH) with obesity and type 2 diabetes is a major determinant factor for the continued rise of NASH-driven HCC. Unfortunately, the mechanisms underlying the progression from NASH to HCC are not well-understood. Steatosis is characterized by the accumulation of different lipid species, and cholesterol has emerged as an important player in NASH development, which has been shown to promote NASH-driven HCC. However, recent findings indicated a tumor suppressor role of cholesterol in liver carcinogenesis and HCC development. Thus, we examined the contribution of hepatic steatosis with or without cholesterol accumulation induced by dietary or genetic approaches in liver tumorigenesis and whether the role of cholesterol in NASH-driven HCC is species-dependent. While diethylnitrosamine (DEN) treatment to rats or mice fed a choline-deficient diet decreased the hepatic steatosis, feeding an atherogenic diet enriched in cholesterol potentiated the liver tumor markers. Similar effects were observed in DEN-treated transgenic SREBP-2 mice but not wild-type (WT) mice fed a regular chow diet. Remarkably, long-term feeding of a high-fat high-cholesterol diet (HFHC) but not a high-fat diet (HFD) to WT mice caused severe NASH with spontaneous progression to HCC. A similar outcome was observed in MUP-uPA transgenic mice fed a HFHC diet, which resulted in increased liver tumors and expression of the genes involved in the immune checkpoints. Ezetimibe treatment ameliorated chronic liver disease and, more importantly, tumor multiplicity in HFHC-fed MUP-uPA mice or DEN-treated WT mice. Thus, these results revealed a differential role of steatosis and cholesterol in NASH-driven HCC and indicated that the tumor-promoter role of cholesterol is species-independent and associated with impaired immunosurveillance.

## 1. Introduction

Hepatocellular carcinoma (HCC) is the ultimate stage of chronic liver disease and a leading cause of cancer-related death [1]. Of all the etiologies leading to HCC development, nonalcoholic steatohepatitis (NASH), an advanced stage of nonalcoholic fatty liver disease (NAFLD), is of utmost concern due to its association with the type 2 diabetes epidemic, positioning NASH as a major cause of the increased HCC cases [2]. Indeed, overweightness and obesity are known risk factors for cancer development, including NASH-driven HCC, which develops in the background of steatosis, liver injury, inflammation, and fibrosis [1,2,3]. Given the metabolic component of fatty liver, the concept of metabolic-associated fatty liver disease (MAFLD) has been recently proposed to better define the metabolic dysfunction that drives the classical NAFLD [4].

Hepatic steatosis is the first stage of MAFLD and is characterized by the accumulation of lipids in hepatocytes. Although this event can be reversible, steatosis can sensitize to multiple secondary hits, leading to NASH development [5]. Several lipid species accumulate during the progression of MAFLD to NASH, including triglycerides, diacylglycerol, free fatty acids, ceramides, and cholesterol, among others, leading to the hypothesis that the type rather than the amount of lipids contributes to the transition from steatosis to NASH [6,7]. In this regard, the increase of hepatic cholesterol levels, including the specific trafficking and accumulation into the mitochondria [8], has emerged as a critical player in NASH development [9]. In line with this transition, cholesterol has also been identified as a tumor promoter in the NASH-to-HCC progression [10,11,12,13,14,15,16]. However, despite this evidence, recent findings have described a tumor-suppressor role of cholesterol in HCC development [17,18]. For example, through the sequestration of CD44 in lipid rafts, cholesterol inhibits HCC invasion and metastasis [19]. Moreover, cholesterol attenuated the progression of DEN-induced HCC in mice by inhibiting the SCAP-mediated fatty acid de novo synthesis [20], and high serum cholesterol levels increase the antitumor function of natural killer cells, leading to a reduction in the growth of liver tumors [21].

Thus, given the controversial role of cholesterol in HCC development, the aim of the present study was to address the following questions: (1) Does hepatic steatosis per se promote NASH-driven HCC? (2) Is the effect of cholesterol in liver tumorigenesis species-dependent? (3) Does cholesterol promote liver tumorigenesis regardless of the mechanism of cholesterol accumulation? Using different dietary and genetic models of hepatic steatosis and cholesterol accumulation in rats and mice, our data indicate that cholesterol rather than steatosis per se promotes NASH-driven HCC.

## 2. Materials and Methods

### 2.1. Animal Models and Treatments

C57BL/6J mice and Sprague–Dawley rats were purchased from Charles River Laboratories. SREBP-2 transgenic mice [22] were purchased from Jackson Laboratories (Strain name: B6; SJL-Tg(rPEPCKSREBF2)788Reh/J). For the induction of tumorigenesis in rats, Sprague–Dawley albino rats with 150 g of weight were treated with diethylnitrosamine (DEN) (13–15 mg/day; 135 mg/L in the drinking water, protected from the light and changed every 2 days), as described previously [23], with parallel groups without DEN treatment. These animals were fed for ten weeks either with a regular chow diet (RD, Teklad 2014) as a control diet, a choline-deficient diet (CD, from Dyets Inc. #D518753), or an atherogenic diet (ATH, regular diet with a custom addition of 1.25% cholesterol and 0.5% cholate) [24]. In the protocols using Tg SREBP-2 animals, mice received DEN in the drinking water (50 mg/L) and were maintained 4 months on a regular diet. For tumorigenesis studies, on postnatal day14 C57Bl/6j mice were injected i.p. with a single dose of DEN (25 mg/kg) [3,25] and, 4 weeks later, were introduced to different diet groups. One cohort of animals was fed RD, CD, or ATH diets for 5 months. Another cohort of animals was fed with a high-fat diet (HFD, containing 60% calories from fat, Research diets Inc. #D12492), HFHC diet (identical to HFD with 60% calories from fat with added 0.5% cholesterol), or HFHC with added Ezetimibe (Ezetrol, Barcelona, Spain) (100 mg/kg of diet, equivalent to 10 mg/Kg/day) for 24 weeks. Another cohort of animals was fed with RD, HFD, or HFHC diets for 10 months without a previous DEN treatment. The experimental design and models used are summarized in Appendix A. Only male animals were used throughout this study, with the ages at the start of the treatments indicated at each experimental procedure. Animals were housed in ambient and sanitary controlled conditions at the University of Barcelona School of Medicine vivarium. All procedures involving the animals and their care were approved by the Ethics Committee of the University of Barcelona and local authorities (Generalitat de Catalunya, Barcelona, Spain) with the numbers 5865 (studies on rats), 5864, 8124, and 9546 (dietary and DEN treatment in mice) and 5909 (Tg SREBP-2 mice), following the national and European guidelines for the maintenance and husbandry of research animals.

### 2.2. Biochemical Measurements

The determination of the ALT, AST, total cholesterol, and triglycerides in the serum and tissue homogenates were performed on a Siemens Advia Chemistry Systems 2400 autoanalyzer, and the total bile acids were measured with a TBA Spinreact kit on a Siemens Dimension Vista autoanalyzer. These measurements were performed by standardized procedures of the Biomedical Diagnostic Center, (Hospital Clínic, Barcelona, Spain). The protein concentrations were measured with the Pierce BCA protein assay kit following the manufacturer’s instructions.

### 2.3. Histological Analysis

Fresh liver samples were fixed in neutral-buffered 10% formalin for 24 h and, afterwards, were dehydrated in 70% ethanol and embedded in paraffin. The tissue blocks were sliced at 7 μm and stained for a general morphological evaluation with the standard hematoxylin and eosin or with Sirius red (Sigma-Merk Life Sciences, Madrid, Spain) for collagen staining and the assessment of fibrosis by Sirius red positive staining quantification by ImageJ free software (NIH). Additionally, the tissue samples were immediately frozen after sacrifice in a cryopreservation medium OCT (Sakura-Finetek, Barcelona, Spain) and stored at −80 °C. Frozen sections of 10–15-μm thickness were stained with Oil Red O (Sigma) to determine the presence of neutral lipid infiltration. To evaluate the presence of non-esterified cholesterol, the frozen sections were fixed with paraformaldehyde 4% for 30 min, stained with filipin (0.2 mg/mL) for 30 min at 4 °C, mounted, and imaged at fluorescence microscope, as described previously [7].

Immunohistochemistry was performed at the IDIBAPS Biobank Histology Core (Barcelona, Spain) on 2-μm-thick tissue sections from paraffin-embedded samples using the Leica Microsystems’ Bond-max™ automated immunostainer together with the Bond Polymer Refine Detection System (Leica Microsystems, Barcelona, Spain). Briefly, after antigen retrieval with a citrate buffer, pH 6 for 20 min, the samples were immunostained with specific primary antibodies (Anti-Alpha fetoprotein Abcam #ab46799; Anti-YAP, Cell Signaling Technologies #14074; Anti-Glul Abcam #ab73593; Anti-Ki67 Abcam #ab15580) developed with diaminobenzidine and counterstained with Harris hematoxylin (PanReac Química, Barcelona, Spain). The IHC and H&E sections were scanned with a 3D Histech Pannoramic DESK II DW digital slide scanner and analyzed with Pannoramic Viewer 1.15 software.

### 2.4. Macroscopic Tumor Quantification

A whole freshly harvested liver was photographed on a glass surface with squared lines spaced 5 mm apart, flipped over on another glass surface, and photographed on the other side. These pictures were used to digitally quantify the area of macroscopic tumors in the liver from the pictures from both sides using ImageJ (NIH). First, the image was calibrated using the underlying 5-mm lines, and afterwards, the positive tumor areas on the liver surface were manually selected and quantified.

### 2.5. Western Blotting

Tissue samples were homogenized in a RIPA lysis buffer supplemented with anti-proteases (Complete, Roche, Madrid, Spain) and antiphosphatases (PhosSTOP, Roche). The samples were rotated 10 min at 4 °C, vortexed, and spun down 5 min at 5000× *g*. All supernatants were collected and quantified for the protein concentration using a BCA reagent (Thermo Scientific Pierce, Waltham, MA, USA). Twenty to eighty micrograms of liver lysates were run in 4–12% SDS–polyacrylamide gel electrophoresis (Criterion, Bio-Rad) and electrotransferred to nitrocellulose membranes (Trans-Blot Turbo, Bio-Rad) following the manufacturer’s instructions. After blocking with 5% BSA, the membranes were incubated with primary antibodies at 4 °C overnight. The following antibodies were used: anti-beta actin-HRP (Sigma, cat#A3854), anti- Cytokeratin 19 (Abcam, Cambridge, UK cat#ab15463), anti-Gp73 (Golm1) (Santa Cruz Biotechnology Dallas, TX, USA (cat#SC48011), and alpha smooth muscle actin (α-SMA, *Acta2*, (Abcam cat#ab5694). After extensive washing and incubation with the corresponding secondary antibodies, the membranes were developed with a Pierce ECL Western Blotting Substrate (Thermo Scientific-Pierce) and imaged with the image-capturing instrument LAS4000 (GE Healthcare, Chicago, IL, USA) and analyzed with ImageJ free software (NIH, Bedestha, MD, USA).

### 2.6. RNA Isolation and RT-PCR

The tissue samples were homogenized using TRIzol reagent (ThermoScientific-Invitrogen, Waltham, MA, USA), phases-separated with chloroform, and the RNA was isolated and further cleaned from the aqueous phase using RNeasy columns (Qiagen, Dusseldorf, Germany). The RNA was quantitated, and its purity assessed using the Nanodrop microvolume spectrophotometer (Thermo Fisher Scientific). Real-time PCR was performed using the iScript™ One-Step RT-PCR Kit with SYBR^®^ Green (Bio-Rad) following the manufacturer’s instructions. The quantification of a given gene, expressed as a relative mRNA level compared with the control, was calculated after normalization to a housekeeping gene that showed no variations in the genotype or treatment. The specificity of the PCR amplification was confirmed by the melting curve analysis ensuring that a single product with its characteristic melting temperature was obtained. Oligonucleotides for the target genes analyzed were designed using the Primer3Plus bioinformatics resource [26] or available oligodatabases (qPrimerDepot) [27]. Oligos for the qPCR sequences are shown in Appendix A.

### 2.7. NAS Scoring

The NAFLD activity score (NAS) was calculated by a registered pathologist unaware of the treatments. The NAS grading system was described as a simple NAFLD scoring system with a high reproducibility that is applicable for different rodent models and for all stages of NAFLD etiology [28]. NAS is calculated as the unweighted sum of the scores for steatosis (including macrovesicular and microvesicular steatosis separately and by hepatocellular hypertrophy) and inflammation (scored by analyzing the amount of inflammatory foci/field).

### 2.8. Determination of Hydroxyproline Levels in Liver Tissue

The hepatic hydroxyproline content was determined as described [29]. In brief, the tissue samples were hydrolyzed in 6-mol/L HCl overnight at 100 °C and purified 4-hydroxy-L-proline standards for 20 min at 120 °C. The free hydroxyproline content from each hydrolysate was oxidized with Chloramine-T. The addition of the Ehrlich reagent resulted in the formation of a chromophore that was read at 550 nm. The data were normalized to the liver wet weight.

### 2.9. Quantification and Statistical Analyses

All data are presented as the means ± SEM. In each experiment, and *n* defines the number of mice. The *t*-test was used to analyze the differences between two groups. To evaluate the differences between more than two groups, a one-way Analysis of Variance (ANOVA) was used with a Bonferroni multiple comparison post-test. The criterion for significance was set at *p* < 0.05. The statistical analyses were performed using GraphPad Prism version 5.

## 3. Results

### 3.1. Differential Role of Steatosis with or without Cholesterol Accumulation in Liver Tumorigenesis

Overweight and obesity are risk factors for HCC development, which is associated with the transition from steatosis to NASH. Thus, we first examined whether hepatic steatosis sensitizes to DEN-induced liver tumorigenesis. As cholesterol is associated with NASH progression, hepatic steatosis was induced with or without cholesterol accumulation by feeding an atherogenic (ATH) diet enriched in cholesterol and supplemented with sodium cholate to prevent the metabolism of cholesterol into bile acids through the classic pathway [24,30] or a choline-deficient (CD) diet, respectively. This approach was first tested in rats following the administration of DEN in the drinking water, changed every two days as described [23], and then fed either a CD or ATH diet. As long-term feeding the ATH diet to DEN-treated rats was not well-tolerated, we used this combination for up to 10 weeks. CD diet feeding caused macrovesicular steatosis, as revealed by Oil Red staining and characterized by an increased liver triglycerides content, while ATH administration resulted in microvesicular steatosis with the accumulation of cholesterol levels in the liver, mostly in a free form, as seen by filipin staining, that, unlike CD feeding, resulted in liver damage (H&E) and serum ALT release (Figure 1A–C). Interestingly, the administration of DEN to either CD or ATH-fed animals attenuated the onset of steatosis with reduced triglycerides levels but potentiated the increase of the cholesterol content and liver damage, as seen by the H&E analyses and serum ALT release (Figure 1B,C). In line with the progressive damage, DEN potentiated hepatic fibrosis in rats fed the ATH diet compared to the CD feeding, as seen by the increase in the expression of *Col1a1*, *Acta2*, and *Tgfβ*, and Sirius red staining, as well as α-SMA (Figure 1D–F). In parallel with these findings, DEN administration increased the expression of HCC and HIF-1α markers in rats fed the ATH diet with respect to CD feeding, including Afp and Vegf-α, as well as the expression of STARD1, which has been recently shown to promote NASH-driven HCC [10] (Figure 1G). Moreover, CK-19, a bona fide HCC stem cell marker [31], increased at the mRNA and protein level with a predominant increase seen by immunohistochemistry in the liver sections of rats treated with DEN and ATH compared to the DEN CD group (Figure 1H). These findings thus revealed a differential effect of steatosis per se vs. NASH characterized by increased cholesterol in DEN-induced liver tumorigenesis.

### 3.2. The Sensitization of Steatosis with Cholesterol Accumulation to DEN-Induced Liver Tumorigenesis Is Species-Independent

To examine whether the preceding findings were species-dependent, we fed mice with CD or ATH diet for 6 months after an initial administration of DEN i.p. at 14 days of age. DEN plus ATH feeding caused microvesicular steatosis with a predominant increase in the cholesterol levels and induced liver damage, as seen by the release of ALT, while DEN plus CD feeding resulted in macrovesicular steatosis with an increase in the liver triglycerides levels with no significant increase in the cholesterol and minimal damage (Figure 2A–C). In addition, DEN plus ATH feeding resulted in hepatic fibrosis, as seen by Sirius red staining and the expression of *Col1a1*, *Acta2*, *Tgfβ*, and higher levels of α-SMA (Figure 2B,D–F) and inflammation with increased levels of *Ccl2*, *Tnfα*, and *Il6* compared to DEN plus CD feeding (Figure 2G). Moreover, while no tumors were found yet in these conditions, increased mRNA levels of the HCC markers, including *Afp, Golm1, Krt19, Birc5, Cd44*, and *Lyve1* (Figure 2H) [31,32,33,34], as well as markers of liver regeneration *Mki67*, *Pcna*, and *Ccne1* (Figure 2H) and the protein levels of Gp-73 and CK-19 (Figure 2I,J), were observed in DEN-treated mice fed the ATH diet compared to DEN CD-fed group. Collectively, the above findings established that NASH characterized by cholesterol accumulation potentiates DEN-induced liver tumorigenesis in both rats and mice.

### 3.3. DEN Potentiates Liver Tumorigenesis in SREBP-2 Transgenic Mice

To further establish that the increase in cholesterol in the ATH model of NASH potentiates DEN-induced tumorigenesis, we used transgenic mice expressing SREBP-2 (Tg SREBP-2), the master transcription factor that regulates cholesterol synthesis in the endoplasmic reticulum (ER). Tg SREBP-2 mice and wild-type (WT) mice were treated with DEN in the drinking water and fed a regular chow diet for 4 months. As seen, the expression of *Srebpf2* and the regulatory enzyme in cholesterol synthesis, *Hmgcr*, increased two-to-three-fold in Tg SREBP-2 mice with respect to WT mice, independently of the DEN treatment (Figure 3A). In parallel with these findings, the total liver cholesterol levels increased in Tg SREBP-2 mice with or without DEN administration (Figure 3B). Filipin staining indicated the increase of free cholesterol levels in Tg SREBP-2 mice compared with WT mice, while Sirius red staining revealed liver fibrosis in Tg SREBP-2 mice that was potentiated by the DEN treatment (Figure 3C,D), in line with the increased expression of fibrosis genes *Col1a1*, *Acta2*, *Tgfβ*, and *Spp1* (Figure 3E). In addition, the constitutive de novo cholesterol synthesis in Tg SREBP-2 resulted in an increased expression of ER stress markers *Pdia4* and *Atf4* (Figure 3F), a characteristic feature associated with chronic liver disease and HCC development [35]. Moreover, despite that there were no macroscopic tumors, we found an increased expression of HCC markers, including *Afp, Golm1, Krt19*, and *Birc5*, in Tg SREBP-2 mice treated with DEN compared to the DEN treatment to WT mice, as well as the proliferation marker *Mki67* (Figure 3G). The protein levels of CK19 and Gp73 in DEN-treated Tg SREBP-2 mice were higher than in WT mice treated with DEN (Figure 3H). Thus, these findings indicate that genetic driven cholesterogenesis potentiates DEN-mediated liver tumorigenesis.

### 3.4. Long-Term Dietary Cholesterol Feeding in WT Mice Induces Spontaneous HCC

Given that the long-term combination of DEN and the ATH diet was not well-tolerated in the above studies, we could not extend this approach to induce the appearance of macroscopic tumors in the liver. Consequently, we next examined whether the increase of the liver cholesterol levels could promote NASH-driven HCC in WT mice. As a high-fat diet (HFD) alone does not induce NASH and DEN plus HFD feeding does not completely model NASH-driven HCC [36], we established a dietary NASH-driven HCC approach by feeding WT mice with a HFD diet supplemented with cholesterol (HFHC) for 10 months [37]. While the body weight gain was similar in HFD or HFHC-fed mice, the liver/body weight ratio significantly increased in HFHC-fed mice (Figure 4A). HFD and HFHC feeding induced macrovesicular steatosis with a similar increase in the liver triglycerides content (Figure 4A,B); however, HFHC feeding significantly increased the liver cholesterol levels compared to HFD-fed mice (Figure 4B). In line with the increase in cholesterol levels, the liver damage was greater in HFHC-fed mice compared to HFD-fed mice, with higher serum ALT levels (Figure 4C). Of relevance, HFHC-fed mice exhibited liver fibrosis, as seen by Sirius red staining (Figure 4B), the increased expression of *Acta2* and *Col1a1*, and the hydroxyproline levels (Figure 4D), accompanied with a higher level of the genes involved in inflammation like *Tnfα* and *Ccl2* (Figure 4E), resulting in a higher NAS score in mice fed a HFHC diet compared to HFD-fed mice (Figure 4F). Consistent with the background of liver injury, inflammation, and fibrosis, HFHC diet feeding increased the liver regeneration, as seen by the increased expression of Ki67 (Figure 4G) and higher expression of Afp (Figure 4H) and increased number of tumor and incidences compared to HFD-fed mice (Figure 4I). Thus, these findings indicate that a long-term dietary increase of liver cholesterol is sufficient to induce NASH progression towards HCC.

### 3.5. HFHC Feeding Exacerbates NASH-Driven HCC

In line with the tumor burden in WT mice after long-term HFHC consumption, we next sought to determine the effect of dietary cholesterol in another established spontaneous NASH-driven HCC animal model. The transgenic MUP-uPA mice fed HFD exhibit classical signs of NASH and progressively develop HCC tumors [35,36]. This model is characterized by endogenous ER stress due to the expression of urokinase plasminogen activator (uPA) within 4–8 weeks of age, leading to cell death and the compensatory proliferation of hepatocytes with a low uPA expression to completely regenerate the liver by 12 weeks [38]. Indeed, these animals at 8 weeks of age show a coarse liver surface, immune infiltration, and mild fibrosis at this young age, which spontaneously resolves over time (32 weeks of age) (Appendix A), in line with the normalization in the expression of the ER stress marker *Ddit3*, serum ALT, and *Srebpf2* and *Col1a1* (Appendix A), as well as the genes involved in inflammation (*Tnfα*); tumor markers (*Afp*, *Birc5*, and *Golm 1*); and the immune checkpoint (*Ctla4*, *Endtpd2*, and *IghA*) (Appendix A). Although previous studies demonstrated that feeding HFD to MUP-uPA results in HCC development [35,36], we next compared the effect of HFHC feeding to MUP-uPA mice compared to HFD in the development of HCC. H&E and Oil Red staining revealed a similar degree of hepatic steatosis in MUP-uPA fed a HFD or HFHC diet for 6 months (Figure 5A), consistent with the similar content of liver triglycerides. The cholesterol levels, however, were higher in MUP-uPA mice fed a HFHC diet both in the serum and liver compared to feeding the HFD diet (Figure 5B). While MUP-uPA mice gained body weights to a similar degree by HFD or HFHC diet feeding (Appendix A), the liver weight gain was higher in HFHC-fed than HFD-fed MUP-uPA mice (Appendix A), and this effect was accompanied by increase release of serum ALT (Appendix A). Moreover, MUP-uPA mice fed a HFHC diet developed a higher degree of fibrosis revealed by Sirius red staining and an increase in the expression of *Col1a1*, *Acta2*, and *Spp1* with respect to HFD feeding (Figure 5C,D) and exhibited a higher expression of the genes involved in inflammation: *Tnfα*, *Il1β*, and *Ccl2* (Figure 5E). Consistent with this background of inflammation and fibrosis, MUP-uPA mice fed the HFHC diet developed liver tumors with an increased tumor multiplicity and maximal area, (Figure 5F,G), and Afp marker in serum (Figure 5H), compared to mice fed the HFD diet. Tumors in the MUP-uPA mice fed the HFHC diet showed heterogeneous features, with different shapes and levels of expressions of HCC markers such as Afp and Yap (Figure 5I). Tumors from MUP-uPA mice fed the HFHC diet exhibited an increased expression of tumor markers *Ly6d, Cd44, Golm1*, and *Birc5*, as well as higher levels of the proliferation marker *Mki67* (Figure 5J). The outcome in the MUP-uPA fed the HFHC diet was paralleled to the findings recently reported in DEN-treated WT mice fed the HFHC compared to HFD feeding [10]. Overall, these findings revealed that NASH-driven HCC is potentiated by the HFHC diet compared to HFD feeding.

### 3.6. Tumor Immunobiology Status of MUP-uPA Mice Fed HFHC Diet

As tumor immunobiology is a critical factor for cancer cell survival [39,40,41], we next examined the status of the immune checkpoint genes related to tumor immunosurveillance. The expression of PD-Ll (*Cd274*) was upregulated in the livers of the HFHC-fed MUP-uPA mice compared to HFD feeding, while the expression of its receptor PD-1 (*Pdcd1*), which is expressed in T cells infiltrated in liver tissue, was not changed (Figure 5K). In line with this outcome, Entpd2 expression, which is involved in the accumulation of suppressor cells in HCC [42], showed a significant upregulation in HFHC-fed MUP-uPA mice compared to HFD feeding (Figure 5K). In addition, the level of *Ctla-4* was upregulated in MUP-uPA mice fed both HFHC and HFD, and a similar trend was found with the IgA-isotype *Igha* [43] expressed by liver resident plasma cells. However, the expression of other T-cell inhibitory receptors, such as Tim-3 (*Havcr2*) or *Lag-3* [41,44], and the cancer immune evasion-related factor *Pak4* [45] remained unchanged in MUP-uPA mice, regardless of the type of diet fed (Figure 5K) in parallel to the unchanged expression of indoleamine 2,3-dioxygenase (Ido1) [46]. In line with the findings in MUP-uPA, WT mice injected with DEN at 14 days of age exhibited induced the expression of the immune checkpoint genes (*Cd274, Pdcd1, Ctla4, Havcr2, Lag3*, and *Entpd2*) driven by HFHC feeding compared to HFD-fed mice (Appendix A). These findings reveal that dietary cholesterol regulates tumor immunobiology and stimulates the expression of genes involved in immune checkpoints to favor a milieu prone to T-cell exhaustion.

### 3.7. Blockade of Dietary Cholesterol Absorption by Ezetimibe Inhibits Liver Tumorigenesis and NASH-Driven HCC

To further establish the tumor-promoter role of cholesterol in NASH-driven HCC, we next tested whether the prevention of dietary cholesterol absorption impacts HCC development. In this paradigm, MUP-uPA mice and DEN-treated WT mice were fed HFHC diet supplemented or not with ezetimibe, which prevents the intestinal absorption of cholesterol by targeting Niemann-Pick C1 Like 1 (NPC1L1) in the intestinal lumen [47]. As seen, the dietary treatment with ezetimibe reduced the serum and liver cholesterol levels in both MUP-uPA and DEN-treated WT mice, without a significant change in the liver triglycerides (Figure 6A–C). The ezetimibe treatment ameliorated a HFHC diet-induced increase in the expression of genes involved in fibrosis, *Col1a1*, *Acta2*, and *Spp1* in MUP-uPA and DEN-treated WT mice (Figure 6D,E). More importantly, ezetimibe markedly reduced the multiplicity and maximal area of liver tumors from MUP-uPA and DEN-treated WT mice fed the HFHC diet (Figure 6F,G). Of interest, the tumor multiplicity in DEN-treated WT mice was higher than in the spontaneous model in MUP-uPA mice, although the maximal area was similar (Figure 6F,G). The expression of the tumor markers in both models was similar, with an increase in the mRNA levels of *Ly6d, Afp, Gpc3, Birc5*, and *Lyve1* being higher in DEN-treated WT mice compared to MUP-uPA, while that of *Cd44* was comparable in both models (Figure 6H–J). Accordingly, the upregulation of the immune checkpoints like *Cd274, Ctla4*, and *Entpd2* induced by HFHC feeding in MUP-uPA mice and DEN-treated WT mice was abrogated by the ezetimibe treatment (Figure 6K,L). Additionally, ezetimibe downregulated the increase in the mRNA levels of *Pdcd1, Havcr2*, and *Lag 3* induced by the HFHC diet in the DEN-treated model but not significantly in the MUP-uPA mice (Figure 6K,L). Overall, ezetimibe prevents the role of cholesterol in promoting liver tumorigenesis and NASH-driven HCC.

## 4. Discussion

NASH-driven HCC development is a major cause of cancer-related death with a prediction to keep rising in the future due to its association with the obesity and type 2 diabetes epidemic. The mechanisms of transition from NASH to HCC are poorly understood, which limits the effective treatment of this major health concern. Hepatic steatosis is the first step of MAFLD and is characterized by the accumulation of different lipid species, which can sensitize to secondary hits towards NASH development. As cholesterol accumulation can be a determining factor contributing to the transition from steatosis to NASH, we first examined whether hepatic steatosis favors liver tumorigenesis. With the development of models of hepatic steatosis with or without cholesterol accumulation, we show that macrovesicular steatosis characterized by increased liver triglyceride levels, but unchanged cholesterol accumulation did not promote DEN-initiated liver carcinogenesis, while the increase in cholesterol content markedly aggravated liver tumorigenesis, lending further support to the existing data that cholesterol plays a tumor promoter role in NASH-driven HCC [10,11,12,13,14,15,16]. In line with this possibility, cholesterol conditions the development of a subtype of HCC with high aggressiveness and poor prognosis [48].

However, despite this growing evidence there have been diverse studies that indicated a tumor suppressor role of cholesterol in HCC progression [18,19,20,21]. Several mechanisms underlined this suppressor role of cholesterol in HCC, including the sequestration of CD44 in lipid rafts or the inhibition fatty acid de novo synthesis [19]. Remarkably, high serum levels of cholesterol have been shown to increase the anti-tumor function of natural killer cells leading to a reduction in the growth of liver tumors [21]. To address this controversy, we first analyzed whether the effect of cholesterol in HCC development could be influenced by the mechanism, leading to increased cholesterol accumulation. In this regard, we used a dietary approach and a genetic model of cholesterogenesis to increase hepatic cholesterol levels and examined the impact on liver tumorigenesis. Feeding an ATH diet enriched in cholesterol and supplemented with cholic acid resulted in a significant increment in cholesterol levels, which aggravated DEN-mediated liver tumorigenesis, in contrast to feeding a CD diet despite causing macrovesicular steatosis. Importantly, the differential effect of feeding the ATH vs. CD diet was observed in both rats and mice, indicating that the ability of cholesterol to foster HCC progression is species-independent. The presence of cholic acid in the ATH diet favors the maintenance of hepatic cholesterol levels by preventing its metabolism to bile acids predominantly through the classic pathway [30], as the mitochondrial alternative mechanism of bile acid generation is insensitive to the feedback inhibition by bile acids [49]. As cholic acid can induce HCC [50], we tested an alternative diet-independent approach to increase hepatic cholesterol levels using the Tg SREBP-2 mice, which exhibit a constitutive induction of SREBP-2, the master regulator of cholesterol synthesis in parallel with enhanced expression of its target *Hmgcr* and free cholesterol content. Treatment of Tg SREBP-2 mice but not WT mice with DEN in the drinking water promoted chronic liver disease, with fibrosis, inflammation, ER stress, and increased expression of HCC markers, thus further ensuring that the tumor promoter role of the ATH diet is not due to the presence of cholic acid but to the higher levels of cholesterol content. An interesting effect in the discrimination between hepatic steatosis per se or the cholesterol accumulation in HCC development is the observation that DEN treatment to CD-fed rats or mice decreased the appearance of macrovesicular steatosis and reduced liver triglycerides content. The underlying mechanism for this striking observation will require extensive further work examining the expression and function of players involved in lipogenesis and/or lipid oxidation.

To further explore the role of cholesterol in HCC development, we fed WT mice a HFHC diet for a prolonged time (10 months) and compared this effect with the impact of feeding an HFD. In line with recent findings [37], long-term HFHC but not HFD feeding caused NASH progressing towards HCC with an increased number of liver tumors and significantly higher incidence of HCC development. Similar findings were observed in a model in which chronic ER stress and overnutrition synergize to cause HCC [35,36]. Compared to HFD-fed MUP-uPA mice feeding the HFHC diet resulted in severe NASH, with fibrosis, inflammation and an increase in the number and size of liver tumors, in line with the outcome in DEN-treated WT mice fed HFHC diet [10]. Quite interestingly, HFHC feeding to MUP-uPA mice or DEN-treated WT mice increased the expression of genes involved in immune checkpoints, indicating that cholesterol impairs immune surveillance. This outcome contrasts with the recent observations that the increment in serum cholesterol potentiates the anti-tumor functions of natural killer cells and reduces growth of HCC in *ApoE^−/−^* and *Ldlr^−/−^* mice [21]. Whether differences in the experimental models used (*ApoE^−/−^* and *Ldlr^−/−^*) or the higher content of cholesterol used (2%) account for the different outcome remains to be determined. Additionally, evidence for the promoter role of nutritional cholesterol in NASH-driven HCC is derived from the beneficial effect of ezetimibe treatment to prevent the intestinal absorption of dietary cholesterol. HFHC diet containing ezetrol significantly decreased liver cholesterol content and NASH-promoted HCC in both DEN-treated WT mice and MUP-uPA mice.

## 5. Conclusions

The present study provides evidence that cholesterol rather than steatosis per se plays a role in NASH-HCC progression, lending further support that the strategies targeting liver cholesterol homeostasis may be of potential relevance in NASH-driven HCC. Besides ezetimibe shown here, recent findings indicated that atorvastatin improved HFHC diet mediated HCC development [50], in line with previous findings in a subcutaneous model of HCC [51]. Future research is warranted to explore whether combination therapy between ezetimibe and statins may be superior for the treatment of HCC.

## Figures and Tables

**Figure 1 cancers-13-04091-f001:**
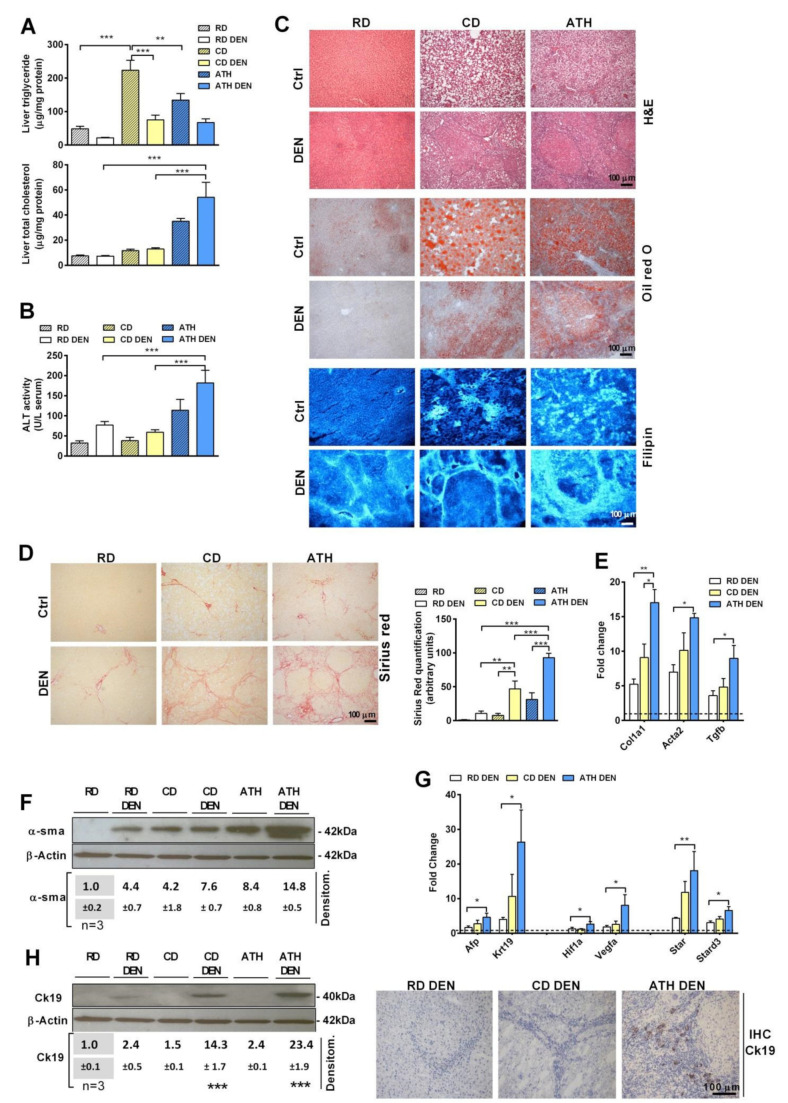
Rats were treated with DEN in the drinking water and fed with RD, CD, or ATH diets for 10 weeks. (**A**) Hepatic triglyceride and total cholesterol composition livers from animals treated with DEN or the vehicle and fed a regular diet (RD), choline-deficient (CD), or a cholesterol-enriched atherogenic diet (ATH) for 10 weeks. (**B**) Transaminase serum levels indicative of liver damage in the animals in the different experimental groups. (**C**) Representative histological staining of liver sections for hematoxylin–eosin (H&E), neutral lipid (Oil Red O), and free cholesterol (filipin). (**D**) Representative histological staining for collagen fibers (Sirius red) and staining quantification of images by ImageJ. (**E**) QPCR determination of fibrosis-related gene expression in the liver. (**F**) Western blot and densitometry of α-SMA in liver tissue, showing significance when compared to their corresponding RD or RD DEN control. Uncropped films from immunoblots in Appendix A. (**G**) QPCR determination of tumorigenesis markers in the livers of DEN-treated animals. (**H**) Representative immunoblot with the densitometry analysis as in (**F**) and representative immunohistochemistry of Ck-19 protein in the livers of animals from different co treated with DEN. Values in qPCR determinations are relative to animals in the RD diet without DEN treatment (value 1.0 depicted as a dotted line) (*N* = 5–7). All values are the mean ± SEM. Symbols *, **, and *** denote statistically significant differences in ANOVA test and Bonferroni post-test (*p* < 0.05, *p* < 0.01, and *p* < 0.001, respectively).

**Figure 2 cancers-13-04091-f002:**
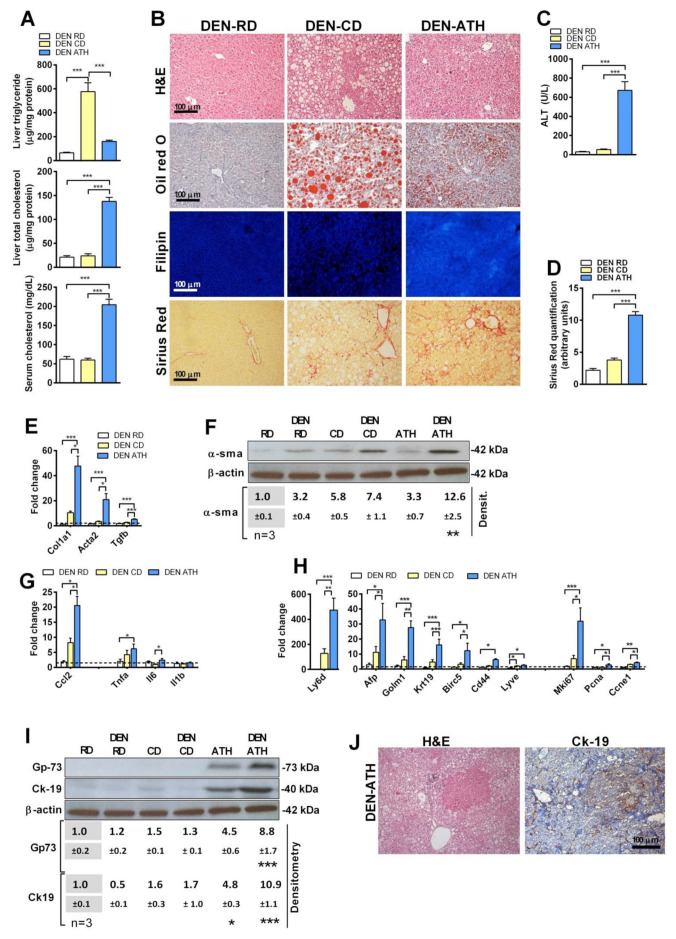
(**A**) Liver triglyceride, liver cholesterol composition, and serum total cholesterol levels in mice treated with a single injection of DEN at 14 days of age and fed with a regular diet (DEN-RD), choline-deficient (DEN-CD), or cholesterol-enriched atherogenic diet (DEN- ATH) for 24 weeks. (**B**) Representative histological staining for hematoxylin–eosin (H&E), neutral lipid (Oil Red O), free cholesterol (filipin), and collagen fibers (Sirius red) of liver sections. (**C**) Transaminase serum levels of DEN-treated animals after 6 months of diet. (**D**) Sirius red staining quantification of liver sections. (**E**) QPCR determination of fibrosis-related gene transcription in DEN-treated animals. Values are relative to the animals in the RD diet without the DEN treatment (value 1.0 depicted as a dotted line). (**F**) Representative immunoblot of fibrosis-marker protein α-smooth muscle actin in liver and a densitometry analysis of the blots relative to the RD group, displaying the significance when compared to their corresponding RD or RD DEN control. Uncropped films from immunoblots in Appendix A. (**G**) Inflammatory markers determined by qPCR, as in (**E**). (**H**) Determination of the HCC and proliferation markers in the liver of DEN-treated animals by qPCR, as in (**E**). (**I**) Immunoblot of Gp73 and Ck-19 proteins in the liver of animals from the different groups and its densitometric analysis. Uncropped films from immunoblots in Appendix A. (**J**) Representative H&E and immunohistochemistry for the CK-19 protein in liver sections of DEN-treated animals. (*N* = 8–10). All values are the mean ± SEM. *, **, and *** symbolize statistically significant differences between the indicated groups (*p* < 0.05, *p* < 0.01, or *p* < 0.001, respectively) in the ANOVA test with Bonferroni’s post-test.

**Figure 3 cancers-13-04091-f003:**
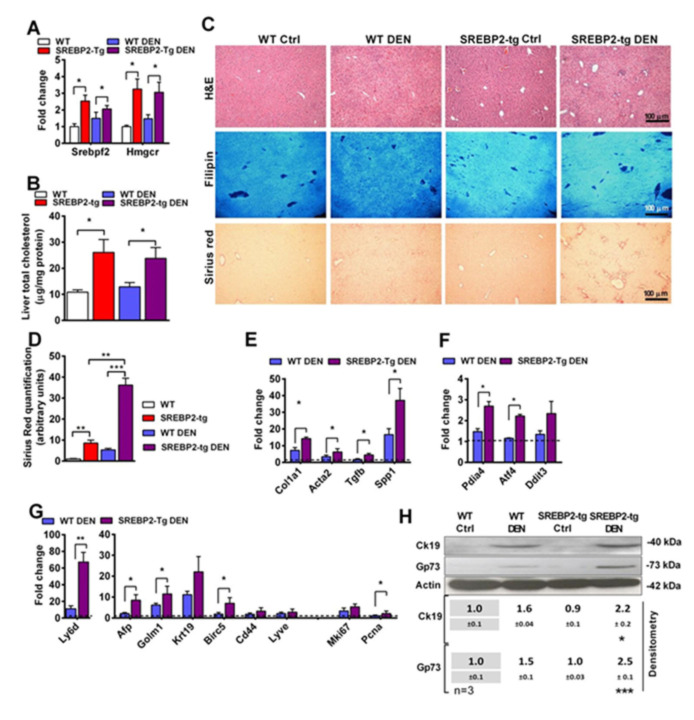
Treatment of wild-type (WT) or transgenic SREBP-2 mice (SREBP2-tg) with DEN in the drinking water for 24 weeks, compared to their respective non-DEN-treated animals (Ctrl). (**A**) Srebpf2 and its target gene Hmgcr mRNA levels in the livers of SREBP2-tg animals, determined by qPCR. (**B**) Total cholesterol levels in the liver tissues in the different groups of animals. (**C**) Representative histological staining of liver sections for hematoxylin–eosin (H&E), free cholesterol (filipin), and fibrosis (Sirius red staining). (**D**) Sirius red quantification from the stained liver sections. (**E**) Determination by qPCR of the fibrosis-related genes (*Col1a1*, *Acta2*, *Tgfβ*, and *Spp1*) of the animals treated with DEN from both genotypes. The values are relative to the WT control. (**F**) Determination by qPCR of ER stress markers *Pdia4*, *Atf4*, and *Ddit3*. (**G**) The expression of the stem cells, proliferation, and HCC-associated markers in WT and SREBP2-tg animals treated with DEN. (**H**) Immunoblotting of Ck19 and Gp73 proteins in the liver of WT and SREBP-2 tg mice and a densitometry analysis of the blots relative to the WT Ctrl group, displaying the significance when compared to their corresponding WT or WT DEN control. Uncropped films from immunoblots in Appendix A. Values in the qPCR determinations are relative to the WT control treatment (1.0 value depicted as a dotted line); only DEN-treated groups are displayed (*N* = 4 to 5 per group). All values are the mean ± SEM. *, **, and *** represent the statistically significant differences (*p* < 0.05, *p* < 0.01, or *p* < 0.001, respectively) in the ANOVA and Bonferroni’s post-test compared to the indicated group.

**Figure 4 cancers-13-04091-f004:**
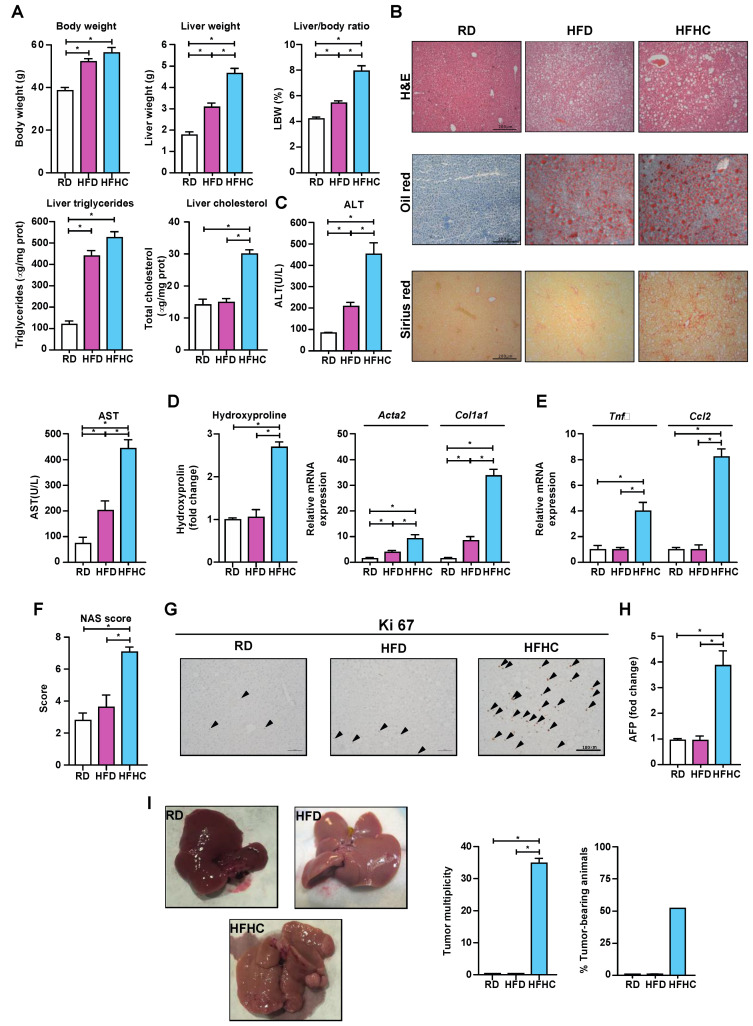
C57BL/6J mice of 6 weeks of age were introduced to RD, HFD, or HFHC feeding without any carcinogenic treatment for 10 months. The number of animals per group: RD (9), HFD (10), and HFHC (10). (**A**) The animal body weight, liver weight, relative liver weight, hepatic triglyceride, and cholesterol composition after 10 months on the specified diets. (**B**) Representative histological staining for hematoxylin–eosin (H&E), neutral lipid (Oil red O), and collagen fibers (Sirius red) of the liver sections. (**C**) Transaminase serum levels (ALT and AST). (**D**) Hepatic tissue hydroxyproline composition and fibrogenesis marker expression (*Acta2* and *Col1a1*) measured by qPCR. (**E**) Liver expression of the inflammation markers (*Tnfα*; and *Ccl2*). Expression levels of the fibrogenesis genes (*Col1a1*, *Acta2*, and *Spp1*). *N* = 6 per group. (**F**) The NAS score of the liver tissue histological preparations. (**G**) Representative images of the immunohistochemistry for the proliferation marker Ki67. (**H**) Serum Afp levels. (**I**) Representative macroscopic whole-liver pictures, tumor number, and incidence of tumors in the different groups of animals. Macroscopic tumors in the liver were quantified from the pictures using ImageJ (NIH), as described in Methods. None of the animals fed RD or HFD developed tumors. All values are the mean ± SEM, symbols * indicates statistically significant differences (*p* < 0.05) by a one-way ANOVA test with Bonferroni’s post-test.

**Figure 5 cancers-13-04091-f005:**
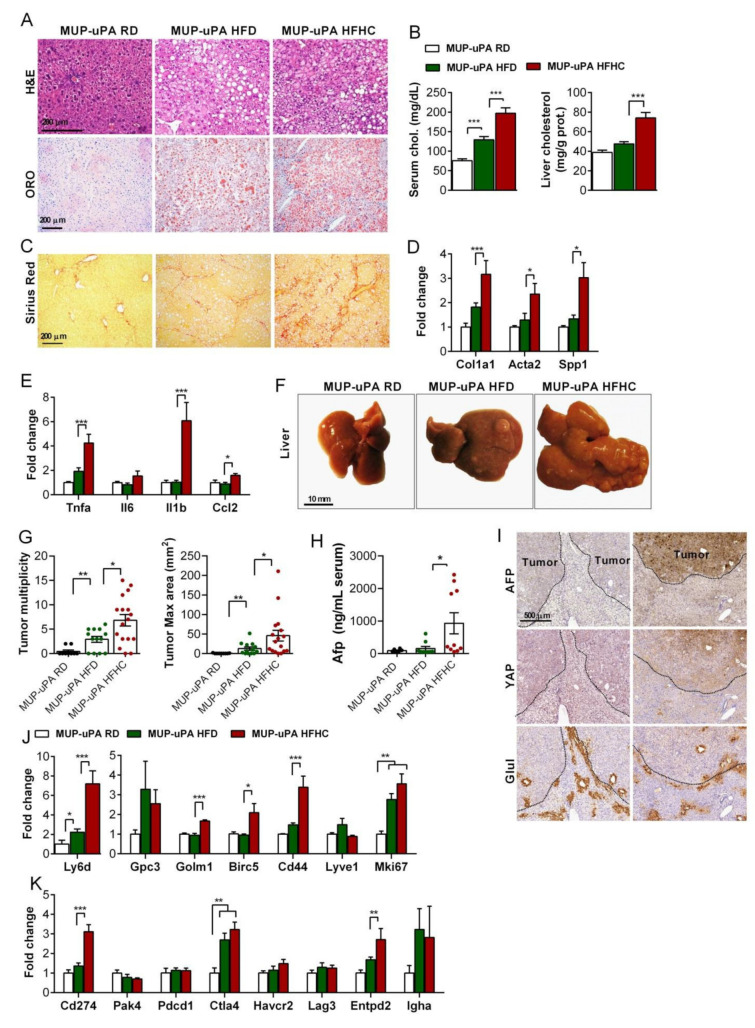
Cholesterol supplementation enhances NASH-associated liver tumorigenesis in MUP-uPA mice. (**A**) Representative histological staining for H&E and neutral lipid (Oil Red O) of the liver sections from MUP-uPA transgenic animals fed a regular diet (RD), high-fat diet (HFD), and high-fat diet high-cholesterol (HFHC). (**B**) Serum total cholesterol concentrations and total cholesterol composition in liver tissue homogenates. (**C**) Histological staining for collagen fibers (Sirius red) of liver sections. (**D**) Transcript qPCR quantification of the fibrogenesis genes (*Col1a1*, *Acta2*, and *Spp1*). *N* = 6 per group. (**E**) Expression levels of several inflammation-related genes (*Tnfα*;, *Il1β*, *Il6*, and *Ccl2*). *N* = 6 per group. (**F**) Representative macroscopic pictures of the whole liver of MUP-uPA transgenic mice in the different diets. (RD, *n* = 9; HFD *n* = 12; HFHC *n* = 17). (**G**) Macroscopic tumor quantification, number, and maximum size (RD, *n* = 9; HFC, *n* = 12; HFHC, *n* = 17). (**H**) Serum Afp levels (*N* = 6–9). (**I**) Immunohistochemical staining for HCC-associated proteins Afp and Yap, and the pericentral hepatocyte marker Glul on consecutive sections of liver tissues with tumors from animals fed the HFHC diet. (**J**) Transcript qPCR quantification of HCC-associated markers (*Afp*, *Birc5*, *Cd44*, and *Ly6d*). *N* = 6 per group. (**K**) Liver mRNA expression of the immune checkpoint and immune inhibitory HCC-associated genes Pd-1L (*Cd274*), Pd-1 (*Pdcd1*), *Ctla-4*, *Havcr2*, *Lag3*, and *Entpd2*). *N* = 6 per group. All values are the mean ± SEM, symbols *, **, *** and **** indicate statistically significant differences (*p* < 0.05, *p* < 0.01, *p* < 0.001 and *p* < 0.0001, respectively) on a one-way ANOVA test with Bonferroni’s post-test.

**Figure 6 cancers-13-04091-f006:**
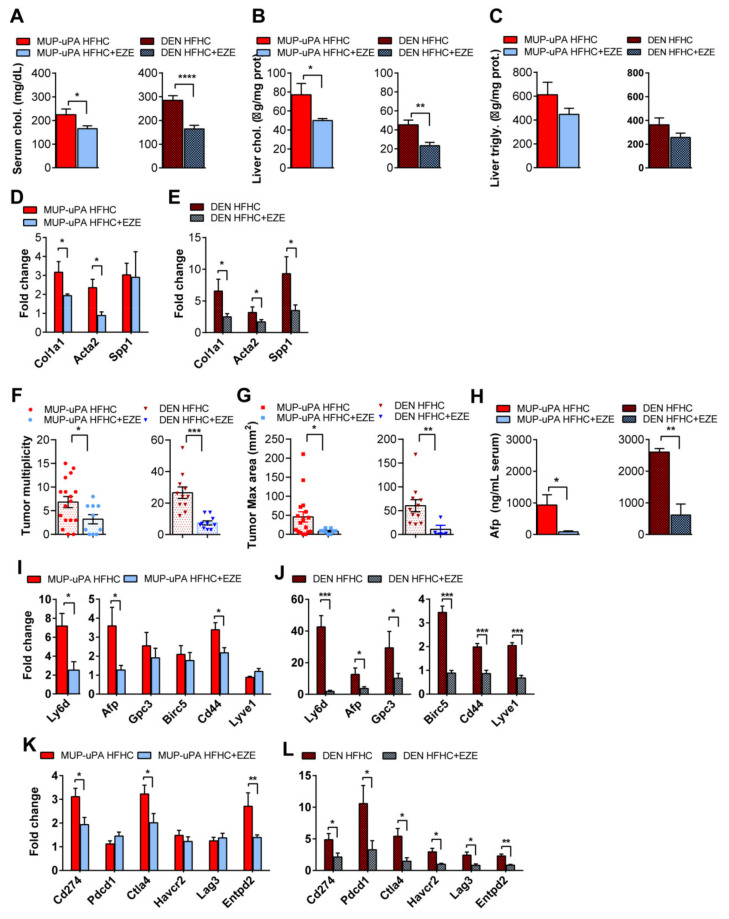
MUP-uPA or DEN-treated animals fed the HFHC diet supplemented with ezetimibe were compared to their corresponding HFHC-fed group. (UPA HFHC *N* = 17, UPA HFHC + EZE = 10; DEN HFHC *n* = 11, DEN HFHC + EZE = 11). (**A**) Serum total cholesterol levels in both models (MUP-uPA and DEN-treated) fed with the ezetimibe-supplemented diet. (**B**) Total cholesterol composition in liver tissue homogenates. (**C**) Triglyceride composition in liver tissue homogenates. (**D**) Expression levels of fibrogenesis genes (*Col1a1*, *Acta2*, and *Spp1*) in MUP-uPA mice. *N* = 6 per group. (**E**) Expression levels of fibrogenesis genes (*Col1a1*, *Acta2*, and *Spp1*). *N* = 6 per group. (**F**) Macroscopic tumor number in both models (MUP-uPA and DEN-treated) fed with the ezetimibe-supplemented diet, compared to their corresponding HFHC-fed group. (**G**) Maximum tumor size. (**H**) Serum Afp levels. (*N* = 6–9 per group). (**I**,**J**) Transcript qPCR quantification of HCC-associated markers (*Ly6d*, *Afp*, *Gpc3*, *Birc5*, *Cd44*, and *Lyve1*) that showed a significant change in either model (MUP-uPA or DEN-treated). *N* = 6 per group. (**K**,**L**) Liver mRNA expression of the immune checkpoint and immune suppression genes Pd-1L (*Cd274*), Pd-1 (*Pdcd1*), *Ctla-4*, *Havcr2*, *Lag3*, and *Entpd2*) in MUP-uPA and DEN-treated mice. *N* = 6 per group. All values are the mean ± SEM, symbols *, **, *** and **** indicate statistically significant differences (*p* < 0.05, *p* < 0.01, *p* < 0.001 and *p* < 0.0001, respectively) on a Student’s *t*-test.

## Data Availability

Data that support the findings are available upon reasonable request. Detailed information on experimental protocols may also be shared on reasonable request.

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
