# Peer review of "Dietary and Genetic Cholesterol Loading Rather Than Steatosis Promotes Liver Tumorigenesis and NASH-Driven HCC"

_cancers, 2021, doi:10.3390/cancers13164091_

Round 1
Reviewer 1 Report
Major comments:
- The H&E results in Figure 1, 2, 3 and 4 are too blurred.
- In result 3.1 and 3.2, the authors fed the rats or mice with CD or ATH diets plus DEN to induce tumorigenesis, however, they haven’t shown tumor development (liver images), i.e., tumor incidence and tumor size. And they haven’t mentioned which part of liver they used for detecting the markers for hepatic fibrosis and tumorgenesis, such as normal liver tissues, tumors or tumor adjacent parenchyma.
- The purpose is to investigate the role of cholesterol other than steatosis in the development of HCC. As DEN-induced tumor development in the absence of NASH, the authors should fed DEN-treated rats or mice with or without cholesterol, to test whether the HCC development was relieved or deteriorated by cholesterol in the absence of NASH. Besides, the authors should also use other tumor inducible models other than DEN, but not induce steatosis, treated with cholesterol to further confirm the role of cholesterol in the HCC development. All the CD, ATH, HFD and HFHC diet could induce the steatosis, are not suitable models for this study, which means all the models used in this study are not suitable.
- In result 3.3, the authors should show the tumor incidence and tumor size between Tg SREBP-2 mice and wild type (WT) mice treated with DEN. The tissues parts for detecting markers are not mentioned, as comment 1. Also, the hepatic triglyceride was increased in Tg SREBP-2 mice, this mice is not a good model for this study.
Minor comments:
α, β need to be corrected.
Author Response
Major comments:
- The H&E results in Figure 1, 2, 3 and 4 are too blurred.
We have modified the H&E images to show a better resolution as requested. These improved images are now shown in revised figures 1-4.
- In result 3.1 and 3.2, the authors fed the rats or mice with CD or ATH diets plus DEN to induce tumorigenesis, however, they haven’t shown tumor development (liver images), i.e., tumor incidence and tumor size. And they haven’t mentioned which part of liver they used for detecting the markers for hepatic fibrosis and tumorigenesis, such as normal liver tissues, tumors or tumor adjacent parenchyma.
We agree with the reviewer´s assessment, but we would like to stress out that the addition of DEN in combination with these diets, particularly the ATH diet, was not well tolerated for a long time as the health of the animals clearly deteriorated beyond 10-12 weeks in rats and 20-22 weeks in mice. Hence given these circumstances, we decided to shorten the combination of DEN and ATH diet for humanitarian reasons. Although this approach was not long enough to see the development of macroscopic tumors in the liver, all tumor markers were already induced both in rats and mice fed the combination of DEN and ATH diet, which suggest the that ATH diet promotes DEN-induced tumorigenesis likely preceding the development of macroscopic tumors. Also, all analyses were performed in normal liver tissues regardless if they had tumors or not unless otherwise stated. We have made appropriate remarks in paragraphs 3.2, 3.3 and beginning of 3.4.
- The purpose is to investigate the role of cholesterol other than steatosis in the development of HCC. As DEN-induced tumor development in the absence of NASH, the authors should fed DEN-treated rats or mice with or without cholesterol, to test whether the HCC development was relieved or deteriorated by cholesterol in the absence of NASH. Besides, the authors should also use other tumor inducible models other than DEN, but not induce steatosis, treated with cholesterol to further confirm the role of cholesterol in the HCC development. All the CD, ATH, HFD and HFHC diet could induce the steatosis, are not suitable models for this study, which means all the models used in this study are not suitable.
The aim of our study was multifold as presented in the Introduction, and one of the major objectives was to examine whether steatosis sensitizes to DEN induced liver tumorigenesis. This is a very important aim as steatosis is the first stage of NAFLD and it can be accompanied or caused by simultaneous accumulation of cholesterol or not. The use of these various dietary approaches, namely CD, ATH, HFD or HFHC diets, allow the induction of a wide spectrum of alterations in the liver with regard to the onset of steatosis with or without cholesterol accumulation. In addition, the degree of steatosis varies from the induction of macrovesicular steatosis and little or not cholesterol accumulation (caused by CD or HFD feeding) to the onset of a microvesicular steatosis with little accumulation of triglycerides and fatty acids, but characterized primarily by the increase of liver cholesterol either by nutritional approach (ATH or HFHC) or genetic approach in the SREBP-2 transgenic mice. The combination of these various possibilities clearly show that the presence of cholesterol regardless of the presence of macrovesicular steatosis emerges as the culprit in the promotion of DEN-induced tumorigenesis and if given for a long term, such in WT mice fed the HFHC diet for 10 months is sufficient to induced macroscopic liver tumors. Thus, we believe our study adds to the conceptual advance that cholesterol is an important factor in the promotion of liver tumorigenesis and cancer in part by impairing immunosurveillance. This outcome dissipates the conundrum centered on the role of cholesterol in NASH-driven HCC clearly supporting its role as a tumor promoter rather than as a tumor suppressor.
- In result 3.3, the authors should show the tumor incidence and tumor size between Tg SREBP-2 mice and wild type (WT) mice treated with DEN. The tissues parts for detecting markers are not mentioned, as comment 1. Also, the hepatic triglyceride was increased in Tg SREBP-2 mice, this mice is not a good model for this study.
This aspect is related to the previous point from the reviewer discussed above (please see answer to point 2). As in the feeding ATH diet in combination with DEN, the administration of DEN to the Tg-SREBP-2 was not well tolerated, precluding extending this approach for a prolonged period of time as to be able to produce macroscopic tumors. Despite the appearance of tumors, there was a clear induction of tumor markers in the DEN-treated Tg SREBP-2 mice but not WT mice.
Minor comments:
α, β need to be corrected.
We have corrected the symbols in the revised version.
Reviewer 2 Report
This is an excellent piece of work which clearly demonstrates that cholesterol rather than steatosis per se plays a role in NASH-HCC progression. Some minor points need to be addressed before acceptance:
-Some figures have blurry panels and size of the image contains an error
-Sirius Red quantification (eg: Image J) should be added
-Fig 4 does not have image size and Ki67 staining cannot be distinguished.
Author Response
This is an excellent piece of work which clearly demonstrates that cholesterol rather than steatosis per se plays a role in NASH-HCC progression. Some minor points need to be addressed before acceptance:
- Some figures have blurry panels and size of the image contains an error
We want to thank the reviewer for bringing this issue. We have increased the quality and improved the resolution of images throughout the manuscript. It appears that this issue stem from the conversion of the images to pdf of the manuscript.
- Sirius Red quantification (eg: Image J) should be added
We have followed this suggestion and have included an additional panel to depict the quantification of the Sirius red when applicable.
- Fig 4 does not have image size and Ki67 staining cannot be distinguished.
We have now included the imagize size as suggested and have improved the quality of the images to clearly show the increase of Ki67 staining in mice fed the HFHC diet.
Reviewer 3 Report
In the manuscript “Dietary and genetic cholesterol loading rather than steatosis promotes liver tumorigenesis and NASH-driven HCC” Ribas et al examined and indicated the important tumor promoter role of cholesterol in NASH- driven HCC.
This is very interesting and well-done paper which significantly advance our fundamental understanding of steatohepatitis. The main idea was strongly supported by different animal models. The results and knowledge generated during the development of this paper will have high potential to impact clinical practice.
Still several points should be addressed:
- I recommend to use new terminology, MAFLD instead on NAFLD (DOI: 1053/j.gastro.2019.11.312)
- The description of the models is a little bit confusing in the material sections. May be authors can make a table with all described models (for supplementary materials).
- The only difference between HFD and HFHC is the concentration of cholesterol? Are the general compositions are similar (composition of FFA for example)? Please add this information to the supplementary section.
- The abbreviation of RD for control group can be confusing for readers. Please modify to “control”. Hence, in Fig. 3 you use ctrl.
- Sirius Red quantifications aremissing.
- Fig 1 "QPCR determination of fibrosis-related gene expression in liver" the control groups are missing. Have you checked aSMA expression? would be great to see if you have the same tendency as in the mouse model.
- Why you changed the DEN protocol in mice in SREBP-2 KO animals? DEN in the drinking water instead of DEN i.p as in Fig.2?
- Fig 2. Have you checked the level of steatosis and fibrosis in mice treated with CD and ATH only?I see the control groups only in aSMA western blot, have you performed ORO and SR stainings in control groups?
- “Protein levels of CK19 and Gp73 in DEN-treated Tg SREBP-2 mice were higher than in WT mice treated with DEN (Figure 3G)”. I don’t see the big differences between the groups in the presented Western blot. Can you perform the densitometric quantification?
- 4 G images have low quality. It is impossible to see any ki-67 positive cells.
- Have you try to apply chow diet supplemented with Ezetimibe? Does it lead to any pathological changes in the liver? Would be great to have this information in supplementary figures or in discussion.
- Fig 1G, please increase the size of the images.
- Animal Ethic committee approval number is missing.
Author Response
In the manuscript “Dietary and genetic cholesterol loading rather than steatosis promotes liver tumorigenesis and NASH-driven HCC” Ribas et al examined and indicated the important tumor promoter role of cholesterol in NASH- driven HCC.
This is very interesting and well-done paper, which significantly advance our fundamental understanding of steatohepatitis. The main idea was strongly supported by different animal models. The results and knowledge generated during the development of this paper will have high potential to impact clinical practice.
Still several points should be addressed:
- I recommend to use new terminology, MAFLD instead on NAFLD (DOI: 1053/j.gastro.2019.11.312)
We have followed this suggestion and have replaced the term NAFLD by MAFLD throughout the manuscript. We have included a brief sentence at the end of the first pagragph in the Introduction to support the use of MAFLD as requested quoting new reference 4.
- The description of the models is a little bit confusing in the material sections. May be authors can make a table with all described models (for supplementary materials).
We believe this is a great suggestion and have included a table (Table S1) in Supplementary Materials to summarize the main features of the different dietary models used in the study.
- The only difference between HFD and HFHC is the concentration of cholesterol? Are the general compositions are similar (composition of FFA for example)? Please add this information to the supplementary section.
This is correct. The HFHC diet is prepared from HFD and added 0.5% cholesterol, and therefore the composition of the HFD with respect to all other nutrients and components is the same. The composition of the commercially available HFD with 60% from fat can be accessed through Research diets:
https://researchdiets.com/formulas/d12492
- The abbreviation of RD for control group can be confusing for readers. Please modify to “control”. Hence, in Fig. 3 you use ctrl.
As described in Methods, RD refers to the group of animals fed the regular chow diet, as shown in Figures 1 and 2. The term control (Ctrl) used in Figure 3 denotes the group of wild type mice not treated with DEN.
- Sirius Red quantifications are missing.
We have followed this suggestion and have included now the quantification of the sirus red staining for Figures 1-3.
- Fig 1 "QPCR determination of fibrosis-related gene expression in liver" the control groups are missing. Have you checked aSMA expression? would be great to see if you have the same tendency as in the mouse model.
As mentioned in the corresponding Figure legends, these values for the control groups are actually shown as dotted lines in the appropriate panels, e. g. Figures 1E and 1G, or Figure 2G and 2H, etc. Given the number of conditions and groups, we prefer to show these values as dotted lines in the respective panels for simplicity.
- Why you changed the DEN protocol in mice in SREBP-2 KO animals? DEN in the drinking water instead of DEN i.p as in Fig.2?
This protocol of DEN administration was described as a model for NASH and HCC induction in rodents, as described (Ref 23). Although presented as Figure 3, the study with the Tg-SREBP-2 mice was simultaneous to the approach we used in rats and both used the same protocol of DEN administration. However, as mentioned the administration of DEN in the drinking water combined with increased cholesterol was not well tolerated and therefore in subsequent studies with mice fed cholesterol enriched diets we prefer to administer DEN as a single injection at 14 days of age. In either way, our data indicate that cholesterol promotes the tumorigenic effect of DEN irrespective of the way of administration (water or IP injection).
- Fig 2. Have you checked the level of steatosis and fibrosis in mice treated with CD and ATH only?I see the control groups only in aSMA western blot, have you performed ORO and SR stainings in control groups?
Indeed these groups were studied but since we present a number of conditions of dietary groups we did not want to overcrowd the figure with additional data, and focused on the groups with DEN induction because our main aim was to check these groups where the tumorigenesis was induced.
The levels of steatosis and fibrosis were similar in mice fed CD and ATH diet regardless of DEN administration.
- “Protein levels of CK19 and Gp73 in DEN-treated Tg SREBP-2 mice were higher than in WT mice treated with DEN (Figure 3G)”. I don’t see the big differences between the groups in the presented Western blot. Can you perform the densitometric quantification?
AS requested by the Editors, we have included the densitometry of all WB, including Figure 3. The increase in Ck19 and in Tg-SREBP-2 mice is statistical significant with respect to WT mice.
- 4 G images have low quality. It is impossible to see any ki-67 positive cells.
We apologize for these images, and have replace them by new improved photos where we show the presence of Ki67 in slices from mice fed HFHC diet.
- Have you try to apply chow diet supplemented with Ezetimibe? Does it lead to any pathological changes in the liver? Would be great to have this information in supplementary figures or in discussion.
This would be an interesting approach to check the effects of ezetimibe in healthy liver. However we only used HFHC diet supplemented with Ezetimibe as our objective was to specifically block cholesterol absorption from the intestine.
- Fig 1G, please increase the size of the images.
The images of Fig 1G (1H in the revised version) have been enlarged to better show the increase of CK19 in DEN-treated mice fed the ATH diet.
- Animal Ethic committee approval number is missing.
The animal ethic committee of the University of Barcelona authorized all the experimentation projects used in this study and were approved by the General Direction of ambiental politics and natural environment of the Local Authority (Generalitat de Catalunya) , with the numbers 5865 (studies on rats), 5864, 8124 and 9546 (dietary and DEN treatment in mice), 5909 (Srebp-Tg mice). We specified these ethic approval numbers in the Method section of the revised version.
Reviewer 4 Report
In this study, Ribas et al. systematically analyze the role of cholesterol in NASH driven cancerogenesis. They convincingly show in different well performed models that cholesterol is a critical promotor of hepatocancerogenesis. Furthermore and interestingly, they reveal that cholesterol also has impact on immune checkpoint expression.
There are only some points I would suggest how to further improve the quality of the manuscript:
- The authors correctly focus in their models on male mice for correct experimental reasons. Still, the authors may discuss potential gender dimorphism based e.g. on available epidemiological data in patients.
- Fig 1a,b: authors may want to check whether there are also significant differences between RD and RD DEN group.
- Authors introduce Golm1, Krt19, Birc5 etc. as HCC markers; I would recommend to provide according references.
- Fig 2e; here, as well as in some other places, symbols are nor printed correctly.
- Fig 4g: quality of images should be improved, eventually accompanied by a numerical quantification/bar graphs.
- Fig 4i: y-axis/tumor multiplicity: here or in legend, it should be explained in more detail what has been counted (surface/histology); x-axis: are RD and HFD actually "zero"? eventually, a sign or an explanation in legend would make this more clear. Furthermore, representative histological images could be included.
- similarly, Fig 5g: y-axis/tumor multiplicity: how has it been determined?
- chapter 3.6. authors describe ... while its receptor PD-1 expressed on T-cells was not changed. Based on mRNA expression in total liver tissue this statement cannot be made. Either it should be confirmed by e.g. IH or IF analysis, or the statement should be adjusted.
- Fig 6f,g: also here, details on the analysis/statistics should be provided in the legend.
Author Response
In this study, Ribas et al. systematically analyze the role of cholesterol in NASH driven cancerogenesis. They convincingly show in different well performed models that cholesterol is a critical promotor of hepatocancerogenesis. Furthermore and interestingly, they reveal that cholesterol also has impact on immune checkpoint expression.
There are only some points I would suggest how to further improve the quality of the manuscript:
- The authors correctly focus in their models on male mice for correct experimental reasons. Still, the authors may discuss potential gender dimorphism based e.g. on available epidemiological data in patients.
Gender differences in HCC susceptibility was not in the scope of this study, although it might be in the aim of future studies, given the gender-specific differences in NASH susceptibility, cholesterol metabolism and HCC incidence.
- Fig 1a,b: authors may want to check whether there are also significant differences between RD and RD DEN group.
Actually these groups, namely mice fed regular diet with or without DEN are shown in Figures 1A and 1B. Although there seems to be some trends for lower triglycerides and increased serum ALT levels, these differences did not reach statistical significance.
- Authors introduce Golm1, Krt19, Birc5 etc. as HCC markers; I would recommend to provide according references.
We have followed this suggestion and have provided references to support the role of these genes as tumor markers:
Golm1 (also known as Gp73) is cited as an HCC associated gene:
Mu X, Español-Suñer R, Mederacke I, Affò S, Manco R, Sempoux C, Lemaigre FP, Adili A, Yuan D, Weber A, Unger K, Heikenwälder M, Leclercq IA, Schwabe RF. Hepatocellular carcinoma originates from hepatocytes and not from the progenitor/biliary compartment. J Clin Invest. 2015 Oct 1;125(10):3891-903. doi: 10.1172/JCI77995. Epub 2015 Sep 8. PMID: 26348897; PMCID: PMC4607132.
Monika Julia Wolf, Arlind Adili, Kira Piotrowitz, Zeinab Abdullah, Yannick Boege, Kerstin Stemmer, Marc Ringelhan, Nicole Simonavicius, Michèle Egger, Dirk Wohlleber, Anna Lorentzen, Claudia Einer, Sabine Schulz, Thomas Clavel, Ulrike Protzer, Christoph Thiele, Hans Zischka, Holger Moch, Matthias Tschöp, Alexei V. Tumanov, Dirk Haller, Kristian Unger, Michael Karin, Manfred Kopf, Percy Knolle, Achim Weber, Mathias Heikenwalder, Metabolic Activation of Intrahepatic CD8+ T Cells and NKT Cells Causes Nonalcoholic Steatohepatitis and Liver Cancer via Cross-Talk with Hepatocytes, Cancer Cell, Volume 26, Issue 4, 2014, Pages 549-564, ISSN 1535-6108, https://doi.org/10.1016/j.ccell.2014.09.003.
Krt19 is among the genes associated to HCC in:
Calderaro, J., G. Couchy, S. Imbeaud, G. Amaddeo, E. Letouze, J. F. Blanc, C. Laurent, Y. Hajji, D. Azoulay, P. Bioulac-Sage, J. C. Nault and J. Zucman-Rossi (2017). "Histological subtypes of hepatocellular carcinoma are related to gene mutations and molecular tumour classification." J Hepatol 67(4): 727-738.
Birc5, also known as survivin is described as marker of HCC.
Llovet JM, Chen Y, Wurmbach E, et al. A molecular signature to discriminate dysplastic nodules from early hepatocellular carcinoma in HCV cirrhosis. Gastroenterology 2006; 131: 1758–67.
60.
- Fig 2e; here, as well as in some other places, symbols are nor printed correctly.
We apologize for these typos, which have been corrected in Figure 2F as well is other images in the revised version.
- Fig 4g: quality of images should be improved, eventually accompanied by a numerical quantification/bar graphs.
As mentioned to the other reviewers, we have improved the quality of the Ki67 staining in Figure 4G.
- Fig 4i: y-axis/tumor multiplicity: here or in legend, it should be explained in more detail what has been counted (surface/histology); x-axis: are RD and HFD actually "zero"? eventually, a sign or an explanation in legend would make this more clear. Furthermore, representative histological images could be included.
Multiplicity of tumors are defined as number of macroscopic tumors visible at the surface of the liver, counted and quantified as described in methods. The number of mice bearing tumors are zero in the RD and HFD group compared to the number shown in Figure 5I. We have clarified this approach in the corresponding figure legend. In addition, some representative histological images have been included.
- similarly, Fig 5g: y-axis/tumor multiplicity: how has it been determined?
Similar to the previous point, the multiplicity of tumors are determined as the number of macroscopic tumors appearing in the liver, which are counted and quantified as described in Methods.
- chapter 3.6. authors describe ... while its receptor PD-1 expressed on T-cells was not changed. Based on mRNA expression in total liver tissue this statement cannot be made. Either it should be confirmed by e.g. IH or IF analysis, or the statement should be adjusted.
We apologize for this confusion. In fact this is correct as the expression of PD-1 is unchanged in the different conditions, and we have reworded the sentence to reflect this fact in the revised version. “….while the expression of its receptor PD-1 (Pdcd1), wihch is expressed on T-cells infiltrated in liver tissue, was not changed (Figure 5K)”.
- Fig 6f,g: also here, details on the analysis/statistics should be provided in the legend.
These details are already provided at the end of the figure legend:
All values are mean ± SEM, symbols *, ** or *** indicate statistically significant differences (p<0.05, p<0.01, p<0.001 respectively) on a Student’s t test.
Round 2
Reviewer 1 Report
I don't have further questions.
The paper could be published in the present form.
Author Response
We want to thank the reviewer for the thorough evaluation of our manuscript, and appreciate the assessment of its acceptance.
Reviewer 3 Report
The authors provided satisfactory answers to all questions, made the necessary changes and significantly improved the manuscript.
1.Hence, the quality of IHC ki67 (4 G) is not sufficient for publication. It is impossible to see ki-67 positive cells, as well as counterstaining. Please provide images of better resolution and magnification, use arrows to mark ki-67 positive cells.
2. "The levels of steatosis and fibrosis were similar in mice fed CD and ATH diet regardless of DEN administration." - has to be mention in the text of manuscript
Author Response
We want to thank the reviewer for the opening remarks about the potential of our manuscript and the valuable suggestions to improve it. Moreover, we appreciate the further comments from the reviewer to further improve the presentation of our findings. Find in the following our point by point reply.
Point 1: we apologize for the quality of Figure 4, which after its incorporation into the word within the manuscript it appears to have lost the quality of the images, in particular the Ki67. We have further improve its appearance in regards of the Ki67 panel and have included a better image incorporating arrows to point out the presence of Ki67.
Point 2: Regarding the suggested addition of the sentence "The levels of steatosis and fibrosis were similar in mice fed CD and ATH diet regardless of DEN administration", it is unclear to us what the reviewer is exactly referring to. We asume the reviewer is referring to Figure 2, in which mice were fed a regular diet (RD), a choline deficient diet (CD) or an atherogenic diet (ATH). Although the steatosis observed by feeding the CD diet is substantially higher (more macrovesicular with higher triglycerides levels) than that observed with the ATH feeding (microvesicular with higher cholesterol content), the fibrosis is much higher with the ATH diet than with the CD diet. This is the case regardless of the DEN administration. Thus, we believe it is not necessary to include the suggested sentence in the text.